# Arabidopsis Hypocotyl Adventitious Root Formation Is Suppressed by ABA Signaling

**DOI:** 10.3390/genes12081141

**Published:** 2021-07-27

**Authors:** Yinwei Zeng, Inge Verstraeten, Hoang Khai Trinh, Thomas Heugebaert, Christian V. Stevens, Irene Garcia-Maquilon, Pedro L. Rodriguez, Steffen Vanneste, Danny Geelen

**Affiliations:** 1Department Plants and Crops, Faculty of Bioscience Engineering, Ghent University, Coupure Links 653, 9000 Ghent, Belgium; Yinwei.Zeng@UGent.be (Y.Z.); inge.verstraeten@ist.ac.at (I.V.); HoangKhai.Trinh@UGent.be (H.K.T.); Steffen.Vanneste@ugent.be (S.V.); 2Institute of Science and Technology (IST) Austria, Am Campus 1, 3400 Klosterneuburg, Austria; 3Department of Green Chemistry and Technology, Faculty of Bioscience Engineering, Ghent University, Coupure Links 653, 9000 Ghent, Belgium; Thomas.Heugebaert@Ugent.be (T.H.); Chris.Stevens@UGent.be (C.V.S.); 4Instituto de Biologia Molecular y Celular de Plantas, Consejo Superior de Investigaciones Cientificas, Universidad Politecnica de Valencia, Avd de los Naranjos, 46022 Valencia, Spain; irene.gmaquilon@gmail.com (I.G.-M.); prodriguez@ibmcp.upv.es (P.L.R.); 5Department of Plant Biotechnology and bioinformatics, Faculty of Sciences, Ghent University, Technologiepark 71, 9052 Ghent, Belgium; 6Lab of Plant Growth Analysis, Ghent University Global Campus, Incheon 21985, Korea

**Keywords:** adventitious roots, abscisic acid, *Arabidopsis thaliana*

## Abstract

Roots are composed of different root types and, in the dicotyledonous Arabidopsis, typically consist of a primary root that branches into lateral roots. Adventitious roots emerge from non-root tissue and are formed upon wounding or other types of abiotic stress. Here, we investigated adventitious root (AR) formation in Arabidopsis hypocotyls under conditions of altered abscisic acid (ABA) signaling. Exogenously applied ABA suppressed AR formation at 0.25 µM or higher doses. AR formation was less sensitive to the synthetic ABA analog pyrabactin (PB). However, PB was a more potent inhibitor at concentrations above 1 µM, suggesting that it was more selective in triggering a root inhibition response. Analysis of a series of phosphonamide and phosphonate pyrabactin analogs suggested that adventitious root formation and lateral root branching are differentially regulated by ABA signaling. ABA biosynthesis and signaling mutants affirmed a general inhibitory role of ABA and point to PYL1 and PYL2 as candidate ABA receptors that regulate AR inhibition.

## 1. Introduction

Abscisic acid (ABA) plays an essential role in controlling the responses of plants to such abiotic stresses as high salinity, low temperature and drought [1,2]. Abiotic stress stimulates ABA synthesis and this triggers a variety of physiological and developmental adaptations [3,4]. Depending on different environmental cues and water availability, ABA regulates, for instance, stomatal transpiration as well as root hydrotropism, and, below ground, it controls root architecture [5,6,7,8]. Modulation of growth is a lasting effect and represents a durable solution for improving, for instance, drought tolerance of crops [9,10]. 

In plants, ABA is synthesized by the cleavage of carotenoids in the plastids involving zeaxanthin epoxidase (ABA1) and nine-cis-epoxycarotenoid dioxygenase (NCED) followed by two oxidation steps (ABA2 and AAO3/ABA3) in the cytoplasm [11]. ABA signaling involves long and short distance transport and perception by PYR/PYL/RCAR (PYRABACTIN RESISTANCE 1/PYR1-like/REGULATORY COMPONENTS OF ABA RECEPTOR) ABA receptors. Upon ABA binding, ABA receptors interact with and inhibit clade A type 2C protein phosphatases (PP2Cs), which are coreceptors in the ternary complex [12,13,14]. Both PYR/PYL/RCARs and clade A PP2Cs are encoded by gene families (14 and 9 members, respectively) and some of them can interact to form receptor complexes with specific, yet overlapping, regulatory functions [5,12,15]. The binding of ABA relieves the PP2C-mediated inhibition of SNF1-related protein kinases (SnRK2) SnRK2.2/2.3/2.6, which are further activated through phosphorylation of Ser171 and Ser175 residues (SnRK2.6 nomenclature) by B2/B3-type RAF kinases [16,17]. Phosphorylation of ion channels [18] and transcription factors by ABA-activated SnRK2s then results in multiple physiological and developmental responses [19,20,21,22,23]. Several transcription factors, such as ABI3, ABI4 and ABI5/ABRE-binding factors (ABFs), have been identified as key regulators of ABA signaling. ABI3 is a B3-type transcription factor, ABI4 is an AP2-type transcription factor and ABI5/ABFs are bZIP-type transcription factors. Only the ABI5 clade of bZIPs have been shown to be directly regulated by SnRK2 phosphorylation [24,25,26], while ABI3 and ABI4 are transcriptionally induced in response to ABA, as a transcriptional target of the SnRK2 target RAV1 transcription factor [27,28]. 

ABA signaling by PYR/PYL/RCARs results in partial functional redundancy, which has precluded the direct identification of the ABA perception mechanism via forward genetic screens for ABA-insensitivity. Detailed analyses of different ABA responses have, nevertheless, led to the discovery of non-redundant functions for specific ABA receptors [29,30,31,32,33]. In contrast to ABA, the ABA signaling agonist, pyrabactin (PB), has a higher degree of selectivity by binding to a subset of ABA receptors [34,35]. This feature enabled the identification of the PYR/PYL/RCAR family of ABA receptors by using a chemical genetic approach for sidestepping genetic redundancy [12,13,14]. Interestingly, PB is an agonist for PYR1 and PYL1, but is an antagonist of PYL2 [35], thus explaining the differential bioactivities of PB and ABA. Subsequent research has led to the development of other synthetic agonists such as quinabactin [36] and opabactin [37] to improve selectivity and activity for certain receptors or enhance potency to trigger certain ABA-mediated biological effects [34,38,39].

Due to its important role in plant physiology and development, much research has focused on understanding the function of ABA in different developmental processes. This has resulted in the uncovering of multiple and complex effects of ABA signaling in the root. One prominent developmental ABA-sensitive growth response of the root is its ability to sense differences in water potential and translate this into growth towards water (hydrotropism). This response was found to depend on SnRK2.2 activity in the cortex [8], and is enhanced in quadruple loss-of-function pp2c mutants [29,40]. Additionally, when roots are not in contact with water as they enter a soil macropore, lateral root initiation is temporally inhibited to avoid developing lateral roots in the air (xerobranching response). This response is associated with PYR/PYL-dependent ABA signaling [7]. Moreover, inhibition of ABA signaling in the endodermis by overexpression of the dominant PP2C allele, *abi1-1D*, rendered lateral root primordium (LRP) development resistant to salt [41]. Consistently, LR growth was less sensitive to ABA inhibition in quadruple in *pyr1pyl1,2,4*, *abi1-1* and *snrk2.2/3/6* mutants [30]. The ABA-induced quiescence of LR primordia was found to depend on *PYL8* and *PYL9*, which control auxin-responsive gene expression in LRPs via direct regulation of MYB77 and MYB44 transcriptional activities, independently of SnRK2-based ABA signaling [42,43].

The above examples indicate that the role of ABA in primary root (PR) growth and lateral root (LR) development is well investigated. In contrast, the role and signaling mechanism underlying ABA regulated adventitious root (AR), which form de novo from non-root tissues, development remains poorly characterized [44]. Here, we specifically focused on determining a first molecular framework for the ABA-regulated AR development after etiolation [45], in the model species *Arabidopsis thaliana*, to clarify the role of ABA during adventitious rooting. To this end, we combined phenotypic analyses in knockout lines impaired in ABA biosynthesis and signaling components and pharmacological applications of exogenous ABA and ABA- signaling agonists. We also developed novel PB analogues, bioisosteres of the sulfonamide moiety, namely phosphonamide and phosphonate substitutions [46]. These analogues could help to tell apart ABA- and pyrabactin-specific action mechanisms as both molecules have a different effect on AR formation. Our analyses unequivocally demonstrate that endogenous ABA levels suppress AR formation, and that this depends on canonical ABA signaling components.

## 2. Materials and Methods

### 2.1. Plant Materials and Growth Conditions

The aba1-1, aba3-1, pyr1-1, pyl1-1 (SALK_054640), pyl2-1 (CSHL_GT2864), pyl3 (SALK_073305), pyl4-1 (SAIL_517_C08), pyl5 (SM_3_3493), pyl8-1 (SAIL_1269_A02), pyl9 (SALK_083621), pyr1pyl1,4 (pyr1 pyl1 pyl4, abbreviated as 114), pyl1,4,5 (pyl1 pyl4 pyl5, abbreviated as 145), pyl1,4,8 (pyl1 pyl4 pyl8, abbreviated as 148), pyl4,5,8 (pyl4 pyl5 pyl8, abbreviated as 458), pyl1,4,5,8 (pyl1 pyl4 pyl5 pyl8, abbreviated as 1458), pyl1,2,4,5,8 (pyl1 pyl2 pyl4 pyl5 pyl8, abbreviated as 12458), pyr1pyl1,2,4,5,8 (pyr1 pyl1 pyl2 pyl4 pyl5 pyl8, abbreviated as 112458), and abi4-1 used in this study are in the Columbia-0 (Col-0) background. The aba2-1, abi1-1, abi2-1 and abi3-1 are in the Landsberg *erecta* (L*er*) background. The single PYR1/PYL/RCAR receptor mutants were obtained from NASC (The Nottingham Arabidopsis Stock Centre). High-order pyl mutants have been reported previously [47]. *Arabidopsis thaliana* plants were incubated following the regime depicted in Appendix A. In brief: Seeds were surface sterilized and sown on 1/2 MS agar vertical plates (1.5 g /L MS, 0.5% (*w*/*v*) Sucrose, 0.05% (*w*/*v*) MES, and 0.8% (*w*/*v*) agar, pH 5.7, described by Trinh et al. (2018) [45]). Plates were incubated for 4 d at 4 °C in the dark for stratification. Plants were germinated by 8 h incubation in the light (22 °C, 70 µmol/m^2^s) before being incubated for 3 d in the dark to induce hypocotyl elongation. Well-elongated seedings were transferred to treatment and further grown on these plates for 10 d in a growth chamber at 70% relative humidity and 22 °C, with 16 h/8 h light/dark cycles (70 µmol/m^2^s). Chemicals were purchased from Sigma-Aldrich. Abscisic acid and pyrabactin were prepared as a 10 mM stock solution in DMSO and stored at −20 °C until use. For dose–response analyses, appropriate volumes were added to autoclaved liquid medium and mixed before being transferred to 12 × 12 cm square petri dishes [45]. Data were collected from at least three independent root growth experiments and analyzed using ANOVA statistics.

### 2.2. General Procedure for the PB Analog Synthesis 

The synthesis of PB analogs was performed as previously described [46]. Diethyl ether and tetrahydrofuran were distilled from sodium benzophenone ketyl or sodium prior to use. Commercially available solvents and reagents were purchased from Sigma-Aldrich or Acros and used without further purification, unless stated otherwise. The purification of reaction mixtures was performed by column chromatography using a glass column filled with silica gel (Acros, particle size 0.035–0.070 mm, pore diameter ca. 6 nm). Solvent systems were determined via initial TLC analysis on glass plates, coated with silica gel (Merck, Kieselgel 60 F254, precoated 0.25 mm). Visualisation of the compounds on TLC plates was performed by UV irradiation or coloration with KMnO4 solution or elemental iodine. High resolution 1H-NMR (300 MHz) and 13C-NMR (75 MHz) spectra were run on a Jeol Eclipse FT 300 spectrometer at room temperature. Peak assignments were obtained with the aid of DEPT, 2D-HSQC and 2D-COSY spectra. The compounds were diluted in deuterated chloroform. Low resolution mass spectra were recorded via direct injection on an Agilent 1100 Series LC/MSD type SL mass spectrometer with electron spray ionisation geometry (ESI 70V) and using a mass selective detector (quadrupole). IR-spectra were obtained from a Perkin-Elmer BX FT-IR spectrometer. All compounds were analyzed in neat form with an ATR (Attenuated Total Reflectance) accessory.

### 2.3. Genotyping

DNA was extracted from individual progeny plants using an extraction buffer containing 100 mM Tris-HCl, 500 mM NaCl, 50 mM ethylenediaminetetraacetic acid (EDTA) and 0.7% sodium dodecyl sulphate (SDS) followed by a precipitation step in isopropanol. Afterwards, a double PCR reaction was carried out using a combination of wild-type primers and T-DNA-specific primers (Appendix A). PCR reactions contained 10 ng DNA, 1 μL of DreamTaq™ Green Buffer, 2.5 pmol forward and reverse primer, 250 μM dNTPs and 0.1 μL of DreamTaq™ DNA Polymerase, adjusted to a 10 μL volume with water. Temperature cycling was 2 min 95 °C, 30 s 95 °C, 30–35 cycles of 30 s 95 °C, 30 s 55 °C, 1 min 72 °C, and 5 min 72  °C. Visualization of the PCR products after gel electrophoresis confirmed the homo- or heterozygosity of the tested plants. Homozygous T-DNA insertion plants were used in further experiments.

### 2.4. Phosphatase Activity Inhibition Assays

Phosphatase activity was assayed using p-nitrophenyl phosphate (pNPP) as a substrate according to Belda-Palazon et al. (2018) [31]. Assays were performed in a 100 μL reaction volume containing 25 mM TRIS–HCl, pH 7.5, 10 mM MgCl2, 1 mM dithiothreitol (DTT), 20 mM pNPP, and 1 μM of recombinant ΔNHAB1 from *Arabidopsis thaliana*. Blank controls without recombinant ABA receptor (PYR1, PYL1) or without pyrabactin (DMSO) were included as a 100% phosphatase activity reference. Pyrabactin and pyrabactin analogs were applied at quantities of 10 µM for the reaction, and the different ABA receptors (PYR1, PYL1) were added at a final concentration of 2 µM. Dephosphorylation of pNPP was monitored with a ViktorX5 reader at 405 nm for 60 min at 30 °C. To calculate PP2C activity, the final absorbance value was subtracted from the initial value and was expressed as percentage of phosphatase activity in the absence of ligand. Three independent experiments were performed, and values are averages ±SD. DMSO blank controls contained 1% (*v*/*v*) DMSO and correspond to 100% of phosphatase activity.

### 2.5. Root Phenotypic Analysis

The macrographs were taken with a Nikon D5000 camera and the images were analyzed with ImageJ^®^. The PR length was measured for individual seedlings via the Segmented Line tool, in which an estimated profile of the root was tracked, and the length of this profile was calculated. The lateral root and adventitious root were scored using a binocular microscope (Olympus, SZX9, Tokyo, Japan).

### 2.6. Seed Germination Assay

*Arabidopsis thaliana* seeds were collected at the same for germination assays. Seeds were surface sterilized and sown on 1/2 MS medium (1.5 g /L MS, 0.5% (*w*/*v*) Sucrose, 0.05% (*w*/*v*) MES, and 0.8% (*w*/*v*) agar, pH 5.7) supplemented with ABA, PB, or PB analogs. Then, plates were stratified at 4 °C for 4 days in darkness. Seeds were germinated at 22 °C under 16h light/8h dark cycles (70 μmol/m^2^s). Germination was scored as a percentage representing the ratio of the number of germinated seeds (emergence of radicles, green cotyledon appearance) to the total seeds initially spread on the plate after 10 days.

### 2.7. Statistical Analyses

Statistical analysis was performed using the GraphPad Prism. One-way ANOVA was performed to determine significant differences between groups of samples, as indicated by different letters. Levels of significance are indicated in the figures by asterisks: * *p* ≤ 0.05; ** *p* ≤ 0.01; *** *p* ≤ 0.001. Values represent the mean of 25–40 Arabidopsis plants in each graph and come from at least 2 independent plant cultures.

## 3. Results and Discussion

### 3.1. Differential Inhibition of AR and LR Formation by ABA and Pyrabactin

To investigate the control of ABA over root development, a series of in vitro experiments were set up to test for primary root growth, lateral root and hypocotyl adventitious root formation (Appendix A). Since ABA has a wide spectrum of activities when exogenously applied, the more selective, synthetic ABA analog pyrabactin was used for comparison [35]. ABA control over root growth is complex as it can both stimulate growth of the primary root and inhibit growth of lateral roots under salt stress conditions by signaling within the endodermal layer [47]. Consistent with the inhibition of primary root growth and cell expansion by exogenous ABA [48], primary root growth, adventitious root (AR) and lateral root (LR) formation were strongly inhibited in a dose-dependent manner (Figure 1). ABA impaired primary root growth at 2.5 µM and above, while at least 10 µM PB was required to show a significant reduction in root length (Figure 1A). At the lowest PB dose (0.25 µM), primary root growth was significantly stimulated, in line with the earlier reports showing that ABA can stimulate growth of the primary root by signaling within the endodermal layer [41,49]. In contrast to the primary root, ABA and PB differentially regulated lateral and adventitious root formation (Figure 1). ABA inhibited LR formation at every concentration applied, whereas PB showed a biphasic dose dependence. Low doses of PB stimulated LR formation and, at 10 µM, the number of LR formed was reduced (Figure 1B). In our assay, etiolated seedlings are transferred to light conditions, stimulating AR formation to, on average, 1.4 AR per seedling (Figure 1C). ABA inhibited AR rooting, which is consistent with the reduction in AR formation in deepwater rice [50] and wheat grown under waterlogged conditions [51]. However, ABA and PB treatments produced a different dose–response curve: low doses of ABA inhibited AR formation more strongly than higher doses (Figure 1C), while PB was a more potent inhibitor, specifically at the higher doses (Figure 1C). The differential responses of three root structures analyzed reveal a complex regulation of root growth and architecture by ABA. Furthermore, it suggests that synthetic analogs of ABA, such as pyrabactin and derivatives, may be used to modulate root architecture to improve, for example, drought tolerance.

In a previous study, we synthesized phosphonamide (E1–E10) and phosphonate (D1–D3) pyrabactin analogues and showed these to reduce the stomatal aperture in tobacco leaf epidermal strips (Appendix A; [45]). These analogues were scored for their impact on germination, root and shoot development. Next to ABA and PB, the compounds E1, E4, E5, E6 and E7 showed activities that were scored as intermediate (++) to strong (+++) (Appendix A). E1 and E5 were somewhat selective, with E1 reducing primary root length and LR number and E5 strongly inhibiting germination, and showing weak activity against primary root length and LR formation. E4, 6 and 7 were more broadly active, causing a more general strong reduction in growth. The effect of the pyrabactin analogues on root and shoot development was quantified, confirming that compounds E4, 6 and 7 strongly inhibited root and shoot growth (Appendix A). Although most compounds showed an inhibitory or no differential growth effect, some compounds stimulated root growth. The compounds E1 and D2 significantly stimulated AR formation and compound E8 stimulated primary root growth (Appendix A). The pharmacological study further confirmed that root growth differentially responds to ABA analogues; however, none of the compounds were sufficiently selective to control only one of the measured root parameters.

To further characterize the putative molecular activity of the pyrabactin analogs, in vitro phosphatase inhibition assays using the ABA receptor proteins AtPYR1 and AtPYL1 and the catalytic core of Arabidopsis HAB1 (ΔNHAB1, amino acids 170–511) were performed. Overall, the activity of the analogs was very weak as compared to PB (Appendix A). E2 reduced PYR1- and E4 PYL1-mediated phosphatase activity, both by about 40%, at a 10 µM application dose. In view of the strong growth inhibition by E1, E5, E6 and E7 phosphonate analogs, these compounds evoke signaling through other PYR receptors than PYR1 and PYL1, or via other signaling routes. Alternatively, in vivo metabolism of these compounds might enhance their activity against PYR1/PYL1.

### 3.2. Defects in the ABA Biosynthesis Pathway Enhance AR Development

Given that exogenous hormone applications may cause artificial physiological responses, mutants with defective ABA synthesis were analyzed. The ABA biosynthesis mutants *aba1-1/zep* (zeaxanthin epoxidase) [52,53,54,55], *aba2-1* (xanthoxin dehydrogenase) [56,57], and *aba3-1* (molybdenum cofactor sulfurase) [58,59] have been well characterized and are defective in different steps of the ABA biosynthesis pathway [11] (Figure 2A). While each of these mutants have reduced ABA levels, only *aba1-1* and *aba2-1* formed significantly more ARs compared to the control plants (Figure 2B). The disruption of ABA biosynthesis leads to the accumulation of carotenoid precursors, which have been shown to stimulate anchor root formation [60]. However, Jia et al. (2019) [60] did not observe differences in anchor root formation in *aba1-1* and *aba3-1*, suggesting that the enhanced AR rooting in *aba1-1* is due to the accumulation of ABA biosynthesis intermediates. The lack of enhanced AR formation in *aba3-1* could reflect non-ABA-related functions of ABA3, because it catalyzes the synthesis of a cofactor also required by other enzymes than AAO3 [61], possibly masking any effects of ABA deficiency on AR formation.

### 3.3. Molecular Framework of ABA Signaling for Suppression of AR Development

Because ABA biosynthesis mutants display a pleiotropic phenotype and may accumulate root modulating intermediates, several mutants, defective in downstream ABA signaling, were analyzed. The perception of ABA is mediated by a ternary ABA signaling complex of clade A PP2C protein phosphatases (e.g., ABI1 and ABI2), PYR/PYL/RCAR ABA receptors (hereafter referred to as PYL), and ABA [12,13,14]. Each subunit of the complex is encoded by a multi-gene family, potentially allowing for high combinatorial complexity [62]. The PYL ABA receptor family in Arabidopsis contains 14 members. As a first attempt to map the individual contribution of members of this gene family, we analyzed single mutants in *PYR1/RCAR11*, *PYL1/RCAR12*, *PYL2/RCAR14*, *PYL3/RCAR13*, *PYL4/RCAR10*, *PYL5/RCAR8*, *PYL8/RCAR3*, and *PYL9/RCAR1*. The mutant *pyl2-1* showed a significant increase in the number of AR formed (Figure 3B), which suggests that ABA signaling through PYL2 might inhibit AR formation. The mutants *pyl1-1* and *pyl8-1* showed a tendency (albeit not significant) for increased AR formation (Appendix A), which aligns with the observed PYL1-mediated phosphatase inhibition by E4, the PB analog that also strongly reduced AR formation (Appendix A), and the role for PYL8 in lateral root formation [42]. Except for *pyl2-1,* other single mutants were not different from control. Given that PB is an antagonist of PYL2, whereas it is an agonist of PYR1 and PYL1, such complexity might contribute to the dose–response curve of PB for AR formation (Figure 1 and Figure 3). Both pentuple and hexuple mutants showed a significant increase in their hypocotyl AR formation numbers, which further suggests that ABA signaling plays an inhibitory role in AR formation and underscores the inhibition of AR formation at the 10 µM ABA dose (Figure 1).

ABA sensing via PYR/PYLs results in the inactivation of clade A PP2Cs, which releases SnRK2 protein kinases [18]. The net output of ABA sensing results in the activation of SnRK2s and downstream activation of transcriptional regulation and physiological responses including ion fluxes. Among the early ABA-regulated transcriptional responses are the *ABI3* (B3 superfamily) and *ABI4* (APETALA2 domain) transcription factors [63,64].

To analyze the putative involvement of PP2C phosphatase activity and TFs in AR formation, we investigated ABA-insensitive *abi1-1*, *abi2-1*, *abi3-1* and *abi4-1* mutants. The *ABI1* and *ABI2* genes encode for two members of the clade A PP2C family and the dominant mutations *abi1-1* and *abi2-1* reduce ABA responsiveness in both seeds and vegetative tissues [65,66]. The *abi3-1* and *abi4-1* mutants contain recessive mutations in ABA-regulated transcription factors [63,64]. Each of these ABA-insensitive mutants showed a significant increase in the number of AR (Figure 3D), in support of a role of ABA in AR inhibition.

## 4. Conclusions

Adventitious root formation in the hypocotyl of Arabidopsis was found to be inhibited by ABA, in part controlled by the ABA receptor PYL1 and PYL2. The signaling route for the ABA inhibition of AR formation differs from the control of LR formation, which relies on PYL8. These results provide additional evidence for differential regulation of the AR and LR root initiation processes.

## 5. Patents

Pyrabactin analogues to modulate plant development. Patent number: 9957288.

## Figures and Tables

**Figure 1 genes-12-01141-f001:**
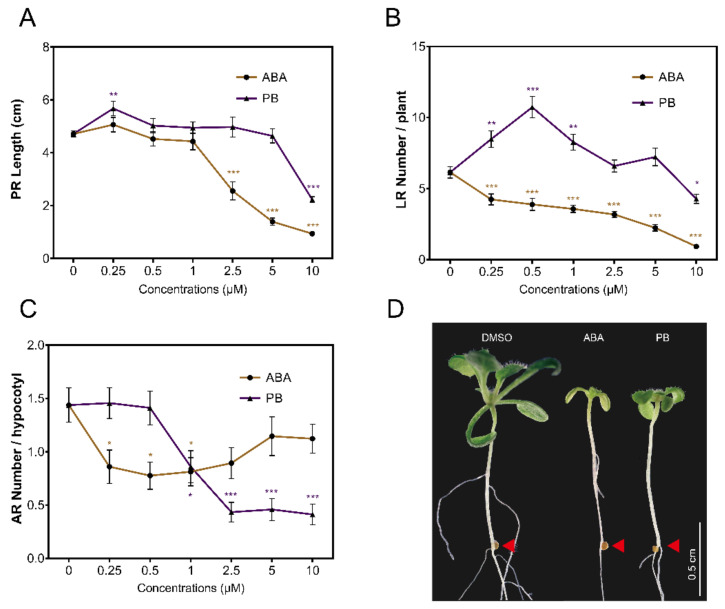
In vitro root development in response to exogenous application of ABA and PB. (**A**–**C**) ABA and PB dose–response curves of primary root growth (**A**), lateral root number per plant (**B**) and adventitious root number per hypocotyl (**C**). ABA and PB was applied at concentrations from 0 µM to 10 µM. (**D**) Adventitious root phenotype of Col-0 plants in the absence or presence of 10 µM ABA and PB; photographs were taken 10 days after de-etiolation; the red triangle points to the hypocotyl root junction. Bar = 0.5 cm. Data are represented as mean values ± se and were obtained from plants grown in three independent experiments (*n* = 20–25). Statistics were calculated through ANOVA and levels of significance are represented as (*) *p* ≤ 0.05, (**) *p* ≤ 0.01, (***) *p* ≤ 0.001.

**Figure 2 genes-12-01141-f002:**
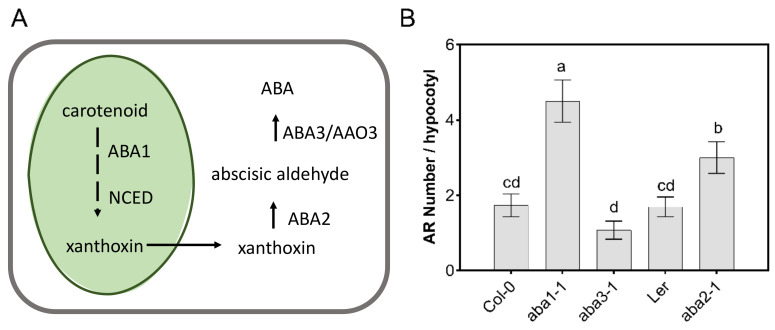
ABA biosynthesis mutants enhance hypocotyl adventitious root formation. (**A**) Scheme of ABA biosynthetic pathway in higher plants with indication of enzymatic conversions affected by mutants that were analyzed. NCED, 9-cis-epoxycarotenoid dioxygenase; AAO, ABA-aldehyde oxidase. (**B**) Adventitious root analysis (number per hypocotyl) in ABA biosynthesis mutants. Recordings were conducted 10 days after de-etiolation. Data, represented as mean values ± se, were obtained from plants grown in three independent experiments (*n* = 20–25). Statistics were calculated using ANOVA, where mean values with different letters are significantly different at *p* ≤ 0.05.

**Figure 3 genes-12-01141-f003:**
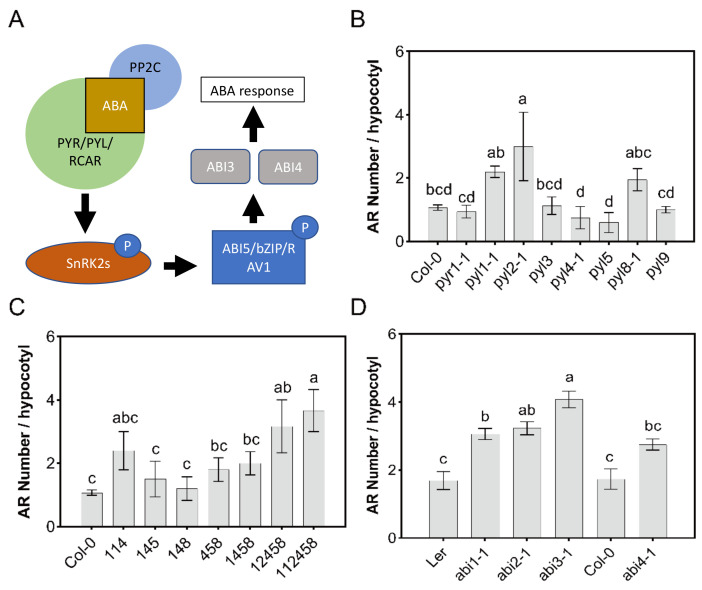
Canonical ABA signaling mutants produce more AR. (**A**) Schematic representation of ABA signaling pathway in plants. ABA binding to PYR/PYLs triggers interaction with PPC2s (such as ABI1 and ABI2). Consequently, SnRK2 is activated to trigger downstream transcriptional and non-transcriptional responses. Phosphorylation of ABI5-clade bZIPs and RAV1 triggers ABA responses, including expression of ABI3, ABI4 and ABI5. PYL, pyrabactin resistance-related; PP2C, protein phosphatase 2C; SnRK2, sucrose nonfermenting-1-related protein kinase 2; ABI, ABA insensitive; P, phosphorylation; RAV1, related to ABI3/VP1; bZIP, basic leucine zipper. The solid line with an arrow indicates direct positive interactions. The solid line with a bar indicates repression. Adventitious root number per hypocotyl for ABA receptor single mutants (**B**), ABA receptor higher order mutants (**C**) and ABA insensitive mutants (**D**) are shown. Plants were grown for 10 days after stratification following 3 days of etiolation in the dark. AR were quantified after 10 days in light. Data, represented as mean values ± se, were obtained from plants grown in three independent experiments (*n* = 20–25). Statistics were calculated through ANOVA, where mean values with different letters are significantly different at *p* ≤ 0.05.

## Data Availability

Raw data supporting reported results are stored on a server in accordance to rules outlined by the data management plan of Ghent University (www.ugent.be/en/research/datamanagement/before-research/datamanagementplan.htm). Accessed on 31 May 2021.

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
