# Peer review of "Arabidopsis Hypocotyl Adventitious Root Formation Is Suppressed by ABA Signaling"

_genes, 2021, doi:10.3390/genes12081141_

Round 1
Reviewer 1 Report
This is an interesting article on the study of the role of ABA in the adventitious root formation in the model plant Arabidopsis. The results show a complex regulation of the growth of various root structures of ABA and its derivatives and can be used to modify the root architecture for practical purposes. There are some comments on the article.
Materials and Methods.
In the M&M section there is no description of the experiment from subsection 3.1 (Fig. 1): doses of ABA and PB, etc.
L.124 – reference for MS medium should be added.
Subsection 2.2. It should be noted that PB analogs (E1-E10, D1-D3) were synthesized earlier (ref. 37).
L.156 - the list of primers and the conditions for the PCR should be indicated.
Results and Discussions.
L. 238 - indicate which compounds are phosphonamide or phosphonate pyrabactin analogs.
Fig. 2 - ZEP is missing in the figure.
L.302. “The mutant pyl1-1 showed a significant increase ...” is probably an error, since the pyl1-1 mutant (ab) differs insignificant from the control (bcd) (Fig. 3B). There is only a significant difference for the pyl2-1 (a).
Supplement.
Fig. S2 - double coding (А-N and E1-10, D1-3) is unnecessary and the names of the compounds, at least the differences between E1-10 and D1-3, should be added.
Fig. S4 - the statistical differences of PB analogs from the control should be added.
L.524-529 - should be deleted (MS Template file).
Author Response
Comments from the Reviewer #1:
This is an interesting article on the study of the role of ABA in the adventitious root formation in the model plant Arabidopsis. The results show a complex regulation of the growth of various root structures of ABA and its derivatives and can be used to modify the root architecture for practical purposes. There are some comments on the article.
Materials and Methods.
- In the M&M section there is no description of the experiment from subsection 3.1 (Fig. 1): doses of ABA and PB, etc.
Reply: the text under section 2.1 was expanded as follows:
2.1 Plant materials and growth conditions.
Abscisic acid and pyrabactin were prepared as a 10mM stock solution in DMSO and stored at -20°C until use. For dose response analyses, appropriate volumes were added to autoclaved liquid medium and mixed [45]. Data were collected from at least three independent root growth experiments and analyzed using ANOVA statistics.
- 124 – reference for MS medium should be added.
Reply: A reference reporting the root phenotype assay and medium preparation was included.
2.1 Plant materials and growth conditions
In brief: Seeds were surface sterilized and sown on 1/2 MS agar vertical plates (1.5 g /L MS, 0.5% [w/v] Sucrose, 0.05% [w/v] MES, and 0.8% [w/v] agar, pH 5.7, described by Trinh et al. (2018) [45]).
- Subsection 2.2. It should be noted that PB analogs (E1-E10, D1-D3) were synthesized earlier (ref. 37).
Reply: reference was included.
2.2 General procedure for the PB analog synthesis
The synthesis of PB analogs was performed as previously described [46].
- 156 - the list of primers and the conditions for the PCR should be indicated.
Reply: The requested information has been added.
2.3 Genotyping
Afterwards, a double PCR reaction was carried out using a combination of wild type primers and T-DNA specific primers (Table S1). PCR reactions contained 10 ng DNA, 1 μl of DreamTaq™ Green Buffer, 2.5 pmol forward and reverse primer, 250 μM dNTPs and 0.1 μl of DreamTaq™ DNA Polymerase, adjusted to a 10 μl volume with water. Temperature cycling was 2 min 95 °C, 30 s 95 °C, 30–35 cycles of 30 s 95 °C, 30 s 55 °C, 1 min 72 °C, and 5 min 72 °C.
Table S1. List of PCR primers.
Name |
Locus tag |
Primer sequences (5’ - 3’) |
pyr1-1_F |
At4g17870 |
TAA AAG CTC GTC GTC GTC TTC |
pyr1-1_R |
GGA AAA GAA AAG GAA AAC CTT TC |
|
pyl1-1_F |
At5g46790 |
ATGGCGAATTCAGAGTCCTCC |
pyl1-1_R |
TTACCTAACCTGAGAAGAGTT |
|
pyl2-1_F |
At2g26040 |
ACCATGGGCTCATCCCCGGCCGTGA |
pyl2-1_R |
TTATTCATCATCATGCATAGGTG |
|
pyl3 _F |
At1g73000 |
AGG AGC AAT TTG AAC TCC CTC |
pyl3_R |
TTG GAA ACC TGG ATT GTT GAC |
|
pyl4-1_F |
At2g38310 |
ACCATGGTTGCCGTTCACCGTCCTT |
pyl4-1_R |
TCACAGAGACATCTTCTTCTTGC |
|
pyl5_F |
At5g05440 |
ATGAGGTCACCGGTGCAACT |
pyl5_R |
TTATTGCCGGTTGGTACTTCGA |
|
pyl8-1 _F |
At5g53160 |
ATGGAAGCTAACGGATTGAG |
pyl8-1_R |
TTAGACTCTCGATTCTGTCGT |
|
pyl9_F |
At1g01360 |
TTC ACT TCA ATG CCC TTG TTC |
pyl9_R |
TAG GTC CCC AAA ACG TCA TAC |
|
LBb1.3 |
|
ATT TTG CCG ATT TCG GAA C |
Results and Discussions.
- 238 - indicate which compounds are phosphonamide or phosphonate pyrabactin analogs.
Reply: Requested information was added.
In a previous study, we synthesized phosphonamide (E1-E10) and phosphonate (D1-D3) pyrabactin analogues and showed these to reduce the stomatal aperture in tobacco leaf epidermal strips (Figure S2;[46]).
- 2 - ZEP is missing in the figure.
Reply: Several steps are not included into the overview figure to focus on those elements that are important for this study.
- 302. “The mutant pyl1-1 showed a significant increase ...” is probably an error, since the pyl1-1 mutant (ab) differs insignificant from the control (bcd) (Fig. 3B). There is only a significant difference for the pyl2-1 (a).
Reply: The text was rewritten to correct for the misrepresentation of the significance of the root formation data.
The mutant pyl2-1 showed a significant increase in the number of AR formed (Figure 3B), which suggests that ABA signaling through PYL2 might inhibit AR formation. The mutants pyl1-1 and pyl8-1 showed a tendency (albeit not significant) for increased AR formation, which aligns with the observed PYL1-mediated phosphatase inhibi-tion by E4, the PB analog that also strongly reduced AR formation (Figure S3), and the role for PYL8 in lateral root formation [43], Except for pyl2-1, other single mutants were not different from control.
Supplement.
- S2 - double coding (А-N and E1-10, D1-3) is unnecessary and the names of the compounds, at least the differences between E1-10 and D1-3, should be added.
Reply: The coding and the names of the compounds was revised.
Figure S2. Chemical structures of phosphonamide (E1-E10) and phosphonate (D1-D3) pyrabactin analogues. E1, Ethyl N‐(pyridin‐2‐ylmethyl)‐P‐(4‐bromonapht‐1‐yl)phosphonamidite; E2, Ethyl N‐(pyridin‐2‐ylmethyl)‐P‐(4‐fluoronapht‐1‐yl)phosphonamidite; E3, Ethyl N‐(pyridin‐3‐ylmethyl)‐P‐(4‐bromonapht‐1‐yl)phosphonamidite; E4, Ethyl N‐(benzyl)‐P‐(4‐bromonapht‐1‐yl)phosphonamidite; E5, Ethyl N‐(pyridin‐3‐ylmethyl)‐P‐(4‐fluoronapht‐1‐yl)phosphonamidite; E6, Ethyl N‐(benzyl)‐P‐(4‐fluoronapht‐1‐yl)phosphonamidite; E7, Ethyl N‐(pyridin‐4‐ylmethyl)‐P‐(4‐bromonapht‐1‐yl)phosphonamidite; E8, Ethyl N‐(pyridin‐2‐ylmethyl)‐P‐(4‐bromophenyl)phosphonamidite; E9, Ethyl N‐(pyridin‐2‐ylmethyl)‐P‐(napht‐1‐yl)phosphonamidite; E10, Hydrogen N‐(pyridin‐2‐ylmethyl)‐P‐(4‐bromonapht‐1‐yl)phosphonamidite; D1, Ethyl N‐(pyridin‐2‐ylmethyl)‐P‐(phenyl)phosphonamidite; D2, Ethyl (pyridin‐2‐ylmethyl) phenylphosphonate; D3, Ethyl N‐(pyridin‐3‐yl)‐P‐(phenyl)phosphonamidite. PB, pyrabactin.
- S4 - the statistical differences of PB analogs from the control should be added.
Reply: A new figure was included displaying the statistics of the phosphatase analyses in the presence of pyrabactin and analogs.
Figure S4. Effect of PB and PB analogs on PP2C phosphatase activity. Phosphatase activity was measured using pNPP as a substrate and PP2C ∆NHAB1 and either PYR1 or PYL1. PB was applied at 10µM and the PB analogs at 1 or 10µM. E2 inhibited PYR1 and E4 inhibited PYL1-mediated ∆NHAB1 phosphatase activity. Data are represented as mean values ± sd were obtained from plants grown in three independent experiments. Statistics were calculated through ANOVA and levels of significance are represented as (*) P ≤ 0.05, (**) P ≤0.01, (***) P ≤ 0.001 and (****) P ≤ 0.0001.
- 524-529 - should be deleted (MS Template file).
Reply: Text was revised.
Reviewer 2 Report
The manuscript “Arabidopsis hypocotyl adventitious root formation is suppressed by ABA signaling” by Zeng et al addresses the effects of ABA (and synthetic analogues) on adventitious root (AR) formation. The finding that ABA inhibits AR formations is not novel and surprisingly the authors ignore this literature in their bibliographic references. Examples of such scientific publications are:
Nguyen et al (2018) Hormonal regulation in adventitious roots and during their emergence under waterlogged conditions in wheat. doi: 10.1093/jxb/ery190.
or
Steffens et al (2006) Interactions between ethylene, gibberellin and abscisic acid regulate emergence and growth rate of adventitious roots in deepwater rice. doi: 10.1007/s00425-005-0111-1
The published research work needs to be properly cited and introduced. Moreover, the results of the present manuscript must be compared and discussed in line with the previous knowledge.
Other comments are:
- In the introduction it is needed a better explanation of what is the definition an AR, It is just briefly described in the abstract.
- Presumably, the hypocotyl elongation treatment is needed for the presence of AR. This need to be clarified. Importantly, ABA effects on hypocotyl elongation needs to be tested for control purposes.
- In the figures, the graphic representations of measurements of root phenotypes (e.g. of the mutants, etc) would improve with images of the Arabidopsis root like it is in Figure 1D. Although, in that figure the seedling on the left (WT/DMSO) is not at the same scale as the other two (WT/ABA and WT/PB), making difficult to analyse the root phenotype.
Author Response
Comments from the Reviewer #2:
The manuscript “Arabidopsis hypocotyl adventitious root formation is suppressed by ABA signaling” by Zeng et al addresses the effects of ABA (and synthetic analogues) on adventitious root (AR) formation. The finding that ABA inhibits AR formations is not novel and surprisingly the authors ignore this literature in their bibliographic references. Examples of such scientific publications are:
Nguyen et al (2018) Hormonal regulation in adventitious roots and during their emergence under waterlogged conditions in wheat. doi: 10.1093/jxb/ery190.
or
Steffens et al (2006) Interactions between ethylene, gibberellin and abscisic acid regulate emergence and growth rate of adventitious roots in deepwater rice. doi: 10.1007/s00425-005-0111-1
The published research work needs to be properly cited and introduced. Moreover, the results of the present manuscript must be compared and discussed in line with the previous knowledge.
Reply: The reviewer is correct to point out that we did not include ABA control of AR in monocotyledonous species. In Arabidopsis, the hypocotyl AR emerge from pericycle like tissue layer which is very different from the origin of AR in grasses. In addition, deep water rice has developed a highly specialized root formation system following the rising water line, quite different from the context in AR are induced in dicots under dry land conditions. Here ABA controls the growth of AR rather than the initiation of new AR. Despite these differences with the root induction system studied here (Arabidopsis), mentioning the role of ABA in monocot rooting can be relevant to the reader and these papers were therefore cited as indicated below.
3.1 Differential inhibition of AR and LR formation by ABA and pyrabactin
ABA inhibited AR rooting, which is consistent with the reduction of AR formation in deepwater rice [51] and wheat grown under waterlogged conditions [52].
Other comments are:
- In the introduction it is needed a better explanation of what is the definition an AR, It is just briefly described in the abstract.
Reply: A description of AR is included.
- Introduction
In contrast, the role and signaling mechanism underlying ABA regulated adventitious root (AR), which form de novo from non-root tissues, development remains poorly characterized [44].
- Presumably, the hypocotyl elongation treatment is needed for the presence of AR. This need to be clarified. Importantly, ABA effects on hypocotyl elongation needs to be tested for control purposes.
Reply: Indeed, etiolation is required, as pointed out in the materials and methods section and in figure S1. The effect of ABA on hypocotyl elongation is not of great relevance in our assays as hypocotyl elongation is nearly complete before ABA and other chemicals are applied. After transfer to the light, there is limited elongation growth of the hypocotyl observed closest to the cotyledons. However, at this region no AR form, which more frequently emerge at the middle or the lower end of the hypocotyl. To minimize possible impact of hypocotyl elongation on the rooting response, we picked etiolated plants showing approximately the same length for the rooting experiment..
- In the figures, the graphic representations of measurements of root phenotypes (e.g. of the mutants, etc) would improve with images of the Arabidopsis root like it is in Figure 1D. Although, in that figure the seedling on the left (WT/DMSO) is not at the same scale as the other two (WT/ABA and WT/PB), making difficult to analyse the root phenotype.
Reply: The image of Arabidopsis shoot part displaying the hypocotyl and adventitious root are displayed at the same scale, which can be verified by looking at the size of the left behind seed coat indicated by a red arrow. Pictures were taken from the plates with plants, but the quality of the images is not sufficient for publication. Since there were no obvious differences in the phenotype of emerged AR, adding the images will not provide valuable information to the paper.
Round 2
Reviewer 2 Report
Thank you for the reply to the comments.
Although the author’s reply to the last one is not satisfactory. If the images of AR were taken without enough resolution for publication, it must be possible to repeat the experiment(s) and do the appropriate imaging. I do not understand the argument is not necessary because “there are no obvious differences” (?!): what about the number of AR in aba1-1 (Fig.2) for example? In Figure 3, again shows mutants with significant differences of AR numbers. The graphs must be accompanied with image examples.
In figure 1D, if the roots are at the same scale, they should be aligned, for instances at the root/hypocotyl junction.
Author Response
We have now included an example of the phenotypes we obtained from which figure 3 was prepared. This is included as a new supplemental figure, Figure S4. The images of the AR phenotype of pyl1-1 and pyl8-1 reveal that the root architecture was very similar to that of Col-0, although there is a tendency (not significant) for producing more AR, as stated in the original text.
Figure 1D was adjusted to position the hypocotyl root junction of all plants at the same level.